# Chemical Language Modeling with Structured State Spaces

**Rıza Özçelik** [1 2]  **Sarah de Ruiter** [1]  **Emanuele Criscuolo** [1]  **Francesca Grisoni** [1 2]

## Abstract

Generative deep learning is reshaping drug design. Chemical language models (CLMs) – which generate molecules in the form of molecular strings – bear particular promise for this endeavor. Here, we introduce a recent deep learning architecture, termed Structured State-Space Sequence (S4) model, into *de novo* drug design. In addition to its unprecedented performance in various fields, S4 has shown remarkable capabilities to learn the global properties of sequences. This aspect is intriguing in chemical language modeling, where complex molecular properties like bioactivity can 'emerge' from separated portions in the molecular string. This observation gives rise to the following question: *Can S4 advance chemical language modeling for de novo design*? To provide an answer, we systematically benchmark S4 with state-of-the-art CLMs on an array of drug discovery tasks, such as the identification of bioactive compounds, and the design of drug-like molecules and natural products. S4 showed a superior capacity to learn complex molecular properties, while at the same time exploring diverse scaffolds. Finally, when applied prospectively to kinase inhibition, S4 designed eight of out ten molecules that were predicted as highly active by molecular dynamics simulations. Taken together, these findings advocate for the introduction of S4 into chemical language modeling – uncovering its untapped potential in the molecular sciences.

## 1. Introduction

Discovering molecules in the vast chemical universe – estimated to comprise up to $10^{60}$ small molecules (Bohacek et al., 1996) – is a 'needle in the haystack problem.' Chemi-cal language models (CLMs) allow exploring this chemical universe efficiently (Skinnider et al., 2021), by enabling the design of desirable molecules without hand-crafted rules (Yuan et al., 2017; Merk et al., 2018a; Grisoni et al., 2021; Ballarotto et al., 2023b; Grisoni, 2023). To achieve this, CLMs represent the molecules as strings that linearize molecular structures (Weininger, 1988) and use the next character prediction task to learn how to design molecules.

The most popular CLM architecture is long short-term memory (LSTM) (Hochreiter & Schmidhuber, 1997; Merk et al., 2018a; Yuan et al., 2017; Grisoni et al., 2021; Segler et al., 2018) networks. While LSTMs have fast generation capabilities, they face challenges in learning global molecular properties due to their information bottleneck (Bahdanau et al., 2015; Gómez-Bombarelli et al., 2018; Chen et al., 2023). Transformers (Vaswani et al., 2017) overcome this bottleneck by multi-head attention (Bagal et al., 2021; Yang et al., 2021), at the cost of increased generation complexity; thereby limiting chemical space exploration. These aspects make it necessary to innovate CLM architectures (Chen et al., 2023).

Structured state-space sequence model (S4) is a recent deep learning architecture with outstanding performance in audio, image, and text generation (Gu et al., 2022). S4 has a 'dual nature': it (a) is trained over the entire input sequences to learn complex global properties and (b) generates strings efficiently – thereby combining some respective strengths of Transformers and LSTMs. Motivated by such *'best of two worlds'* behavior, here we ask the following question: Can S4 advance the current state-of-the-art in chemical language modeling? We find evidence that it can.

Here, we apply S4 to chemical language modeling on SMILES strings and benchmark it on various tasks relevant to drug design – from learning bioactivity to chemical space exploration. Moreover, we further corroborate the promise of S4 via the prospective *de novo* design of kinase inhibitors, validated using molecular dynamics simulations. Our results show the promise of S4 for chemical language modeling, especially in capturing bioactivity and complex molecular properties. To the best of our knowledge, this is the first time that state space models have been applied to molecular tasks, and we expect their relevance for chemical language modeling to increase in the future.

[1]Institute for Complex Molecular Systems and Dept. Biomedical Engineering, Eindhoven University of Technology, Eindhoven, Netherlands. [2]Centre for Living Technologies, Alliance TU/e, WUR, UU, UMC Utrecht, Netherlands.. Correspondence to: Francesca Grisoni <f.grisoni@tue.nl>.

*Accepted at the 1st Machine Learning for Life and Material Sciences Workshop at ICML 2024.* Copyright 2024 by the author(s).

## 2. Structured state-space sequence model (S4)

S4s are an extension of discrete state-space models, widely adopted in control engineering (Hamilton, 1994). Discrete state-space models map an input sequence $u$ to an output sequence $y$, via the learnable parameters $\overline{A} \in \mathbb{R}^{N \times N}$, $\overline{B} \in \mathbb{R}^{N \times 1}$, $\overline{C} \in \mathbb{R}^{1 \times N}$, and $\overline{D} \in \mathbb{R}^{1 \times 1}$, as follows:

$$x_k = \overline{A} x_{k-1} + \overline{B} u_k$$
$$y_k = \overline{C} x_k \quad + \overline{D} u_k \qquad (1)$$

Discrete state-space models define a *linear recurrence*: at $k$-th step, the $k$-th element of the input sequence $u_k$ is fed into the model and used to update the hidden state $x_k$ and to generate an output, $y_k$. $\overline{A}, \overline{B}, \overline{C}$, and $\overline{D}$ control how the input and the hidden state are combined to provide an output.

Besides their recurrent formulation, discrete state-space models can be formulated as a *convolution* with the same set of parameters. By 'unrolling' the linear recurrence (equation (1)), the output sequence $y$ can be obtained via a learnable convolution over the input sequence $u$:

$$y = u * \overline{K}, \quad \overline{K} = f(\overline{A}, \overline{B}, \overline{C}). \qquad (2)$$

This *convolutional representation* reveals a key aspect of state-space models: they can learn explicitly from the entire sequence (via a global convolution filter $\overline{K}$) while preserving recurrent generation capabilites.

S4 models expand upon 'vanilla' discrete state-space models to tackle vanishing gradient issues and high computational costs by introducing additional structure to $\overline{A}$ and $\overline{B}$ (Gu et al., 2020). This structure has made S4s reach state-of-the-art in several generative tasks by a large margin (Gu et al., 2020; 2021; 2022).

## 3. Results and Discussion

We evaluated S4 for its ability to learn from and generate drug-like molecules in an array of tasks, and in terms of multiple molecular properties. LSTMs and Generative Pretrained Transformers (GPTs) were used as benchmarks, since they are the *de facto* approaches in chemical language modeling for *de novo* design (Skinnider et al., 2021; Flam-Shepherd et al., 2022; Grisoni, 2023; Chen et al., 2023). Furthermore, LSTM (recurrent training and generation) and GPT (holistic training and generation) constitute the ideal benchmarks for S4, due to S4's dual formulation (convolution during training and recurrence during generation), which allows inspecting the effect of each of these aspects on the overall performance. Finally, the prospective *de novo* design of putative MAPK1 inhibitors, corroborated by molecular dynamics simulations, was performed to test the potential of S4 in real-world drug discovery scenarios.

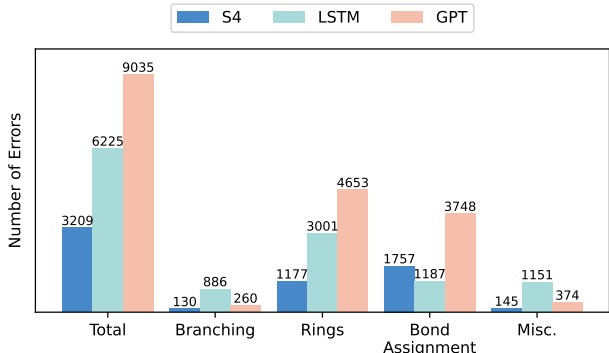

*Figure 1. SMILES design errors, grouped by category and CLM architecture.* Each CLM trained on ChEMBL was used to design 102,400 SMILES strings and the invalid designs are categorized per error. The values indicate the number of errors in each category.

### 3.1. Designing drug-like molecules

S4 was analyzed for its ability to design drug-like small molecules (SMILES length lower than 100 tokens) extracted from ChEMBL database (Gaulton et al., 2017), by focusing on its ability to (a) learn the chemical syntax, (b) capture structural features relevant for bioactivity, and (c) designing structurally diverse molecules.

#### 3.1.1. LEARNING THE SMILES SYNTAX

All investigated CLMs were trained on 1.9M canonical SMILES strings extracted from ChEMBLv31. The generated strings were evaluated according to their (a) *validity*, *i.e.*, the number (and frequency) of SMILES corresponding to chemically valid molecules; (b) *uniqueness*, which captures the number (and frequency) of structurally-unique molecules among the designs; and (c) *novelty*, corresponding to the number (and frequency) of unique and valid designs that are not included in the training set. A high number of 'chemically-valid' designs suggests that the model has learned how to generate plausible molecules, while high uniqueness and novelty values indicate little redundancy among the designs and with the training set, respectively. Although these metrics are vulnerable to trivial baselines (Renz et al., 2019), they provide insights into a model's capacity to learn the SMILES 'syntax'.

All CLMs generated more than 91% valid, 91% unique and 81% novel molecules (Table S1) and their designs approximated the training and test sets in terms of selected molecular properties (Figure S1). These results agree with the literature on CLMs (Brown et al., 2019; Skinnider et al., 2021), and demonstrate the robustness of the model training procedure. S4 designs the most valid, unique, and novel molecules, by generating more novel molecules than the

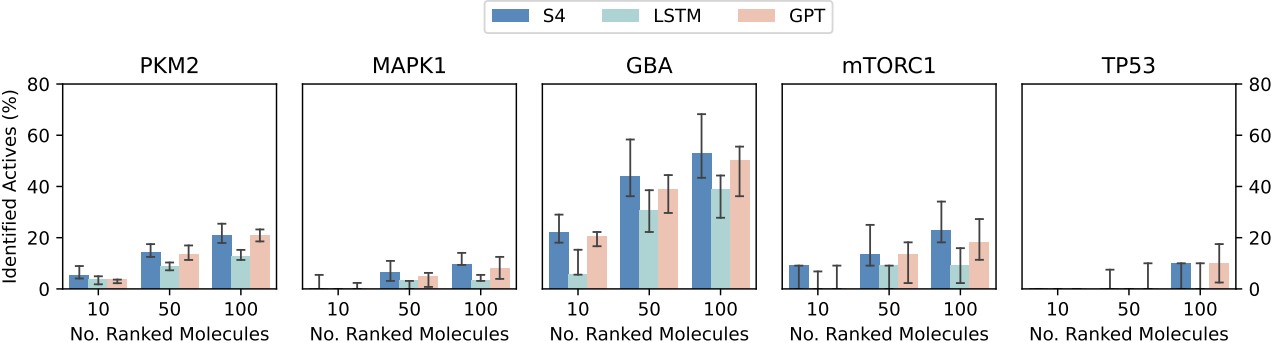

Figure 2. *Retrospective enrichment analysis for all models across five selected macromolecular targets.* The fine-tuned models were used to rank the held-out actives and inactives of the respective protein targets. The percentage of known actives ranked per considered number of test set molecules (10, 50, 100) was computed across ten runs. Bar heights report the median across runs and error bars report the first and third quartiles.

benchmarks (from approximately 4,000 to 12,000 more), and displays a good ability to learn the 'chemical syntax' of SMILES strings.

To shed additional light on the strengths and limitations of S4 in comparison with the benchmarks, we analyzed the sources of invalid molecule generation for all methods in terms of branching and ring errors, erroneous bond assignment, and other (miscellaneous) syntax issues (Figure 1). Interestingly, each method seems to show different types of errors leading to SMILES invalidity. LSTM struggles the most with branching, and performs the best with bond assignment, while GPT struggles the most with rings and bond assignment, and has intermediate performance otherwise. S4 struggles more than LSTM with bond assignment, and generates remarkably fewer errors than both benchmarks in branching and ring design. Our hypothesis is that bond assignment indicates good learning of 'short-range' dependencies, while branching and ring opening and closure require better capturing of the 'long-range' relationships. This suggests that S4 captures long-distance relationships well, in agreement with existing evidence in other domains (Gu et al., 2020; 2021; 2022).

### 3.1.2. Capturing bioactivity

We evaluated S4 for its ability to learn elements of bioactivity. With CLMs this is often achieved with transfer learning (Weiss et al., 2016), which allows transferring knowledge acquired from one task to another task with fewer available data. Via transfer learning, after pre-training a CLM on a large corpus of SMILES strings, the model can be then 'fine-tuned' on a smaller, and task-focused set (*e.g.*, bioactive molecules) by additional training (Segler et al., 2018). Here, we performed five fine-tuning campaigns, focusing on distinct macromolecular targets from the LIT-PCBA (Tran-

Nguyen et al., 2020) dataset: (1) pyruvate kinase muscle isoform 2 (PKM2), (2) mitogen-activated protein kinase 1 (MAPK1), (3) glucocerebrosidase (GBA), (4) mechanistic target of rapamycin (mTORC1), and (5) cellular tumor antigen p53 (TP53).

Evaluating the bioactivity of *de novo* designs (besides synthesis and wet-lab testing) is non-trivial, since this property cannot be fully captured by traditional molecular descriptors, and might not be accurately predicted by quantitative structure-activity relationship models (van Tilborg et al., 2022; Weng et al., 2024). Hence, we used experimentally-tested molecules to evaluate the capacity of a CLM to learn elements of bioactivity retrospectively. Several studies have shown that the likelihoods learned by a CLM during fine-tuning can be used to prioritize designs with high chances of being bioactive (Laban et al., 2022; Moret et al., 2021; Ballarotto et al., 2023a). Based on the same principle, here we used the likelihoods learned by the CLMs to rank existing molecules and evaluate their capacity to prioritize bioactive compounds over inactive ones.

For each of the selected targets, bioactive molecules were used for fine-tuning, with ten random training-validation-test splits. After fine-tuning the CLMs on each target, for each training-test split, we proceeded as follows:

1. With each fine-tuned model and per each target, we predicted the likelihoods of the SMILES strings in the respective test set. The considered test sets resemble a real-world scenario in terms of hit-rate, and they comprise 9 (mTORC) to 54 active molecules (PKM2) and 10,240 inactive molecules (except for TP53, containing 3,301 inactive molecules, Table S2);

2. We ranked the molecules of the test set according to the predicted likelihoods;

3. For each target and each test set, we computed the fraction of actives ranked among the top 10, top 50, and top 100 molecules. The higher the number of active molecules ranked in early portions of the test set by a CLM, the better the model has learned what is relevant for bioactivity on the investigated target.

Our results show variable performance depending on the target (Figure 2). The most challenging target is TP53, on which no model could consistently retrieve actives among the top 10 scoring molecules. Notably, this target has the most challenging test set, where inactive molecules are similar to the actives of both the training and the test sets (Figure S2), potentially indicating the presence of activity cliffs (Maggiora, 2006). MAPK1 and mTORC1 also challenge the CLMs; here, S4 retrieved more active molecules than the benchmarks, especially in the early portions of the test set. PKM2 and GBA are the easiest datasets; here, all CLMs identified bioactive molecules in their top 10, with S4 achieving the highest median across the board. A Wilcoxon signed-ranked test (Woolson, 2007) on the pooled scores across datasets supports the superior performance of S4 compared to the benchmarks ($p < 0.05$), and of GPT compared to LSTM ($p < 0.05$).

Under the constraints of the study design, these results indicate that processing the input SMILES 'holistically' (as GPT and S4 do) leads to capturing complex properties like bioactivity better, with a better performance obtained by S4.

### 3.1.3. CHEMICAL SPACE EXPLORATION

We analyzed the ability of S4 to explore the chemical space, in terms of generating structurally diverse and bioactive molecules. To this end, we employed a commonly-used strategy with CLMs, that is, varying the sampling temperature ($T$) to control chemical diversity (Moret et al., 2020). $T$ affects which elements of a string are generated by a weighted random sampling. When $T \to 0$ the most likely element (based on the CLM prediction) is selected as the next element of the sequence, while the higher the $T$, the more random the selections. $T = 1$ corresponds to using the CLM predictions as the sampling probability of each element at each generation step.

We experimented with an increasing sampling temperatures (from $T = 1.0$ to $T = 2.0$ with a step of 0.25). Each $T$ value was used to generate 10,240 SMILES strings per model across the five targets and all training-test splits. Then, we evaluated the designs based on three metrics (Figure 3):

- *The validity* of the generated strings, which captures how robust the model is to increasing degrees of randomness in preserving a correct syntax.

- *Rediscovery rate*. *De novo* design models are of-

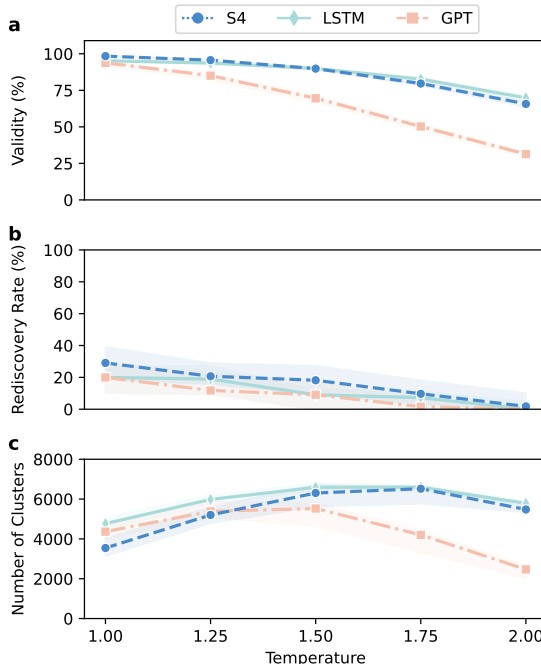

*Figure 3. Model performance when varying the temperature value.*
**(a)** Analysis of the SMILES validity across temperature. **(b)** Variation of rediscovery rate. The models were evaluated for their capability to rediscover bioactive molecules not used for model training or design molecules similar in structure (with a Tanimoto similarity of extended connectivity fingerprints higher than 60%). **(c)** Analysis of the number of diverse groups of scaffolds generated per method. Scaffolds were clustered together if they had a Tanimoto similarity (computed on extended connectivity fingerprints) larger than 60%. For each plot, the solid line indicates the median obtained across the five analyzed protein targets (PKM2, MAPK1, mTORC1, and TP53) with ten runs each, and the shaded area indicates the inter-quartile range. The statistics per individual target can be found in Figure S4.

ten evaluated for their capacity to reproduce existing molecules with experimentally verified biological activities (Brown et al., 2019). Here, to 'relax' the criterion of rediscovery, we considered held-out actives with substructure similarity higher than 60% to a *de novo* design (Tanimoto similarity on extended connectivity fingerprints (Rogers & Hahn, 2010)) to compute rediscovery. Higher rediscovery rates in increased temperatures indicate that the model can explore regions related to bioactivity despite increased randomness.

- *Scaffold diversity*. Designing molecules with novel scaffolds bears relevance in lead identification (Schneider et al., 2006), and can be used as a proxy to evaluate CLMs (Polykovskiy et al., 2020). Here, the novel designs were hierarchically clustered based on their scaffold similarity, to group designs with similar Bemis-

Table 1. *Natural product design with CLMs.* The models were trained on 32,360 natural product SMILES strings from the COCONUT database (Sorokina et al., 2021) and used to generate 102,400 SMILES strings *de novo*. The number and fraction of valid, unique, and novel molecular designs are calculated for each model. Mean and standard deviation of designs' (a) natural-product-likeness (Ertl et al., 2008), (b) the number of $sp^3$ carbons, (c) molecular weight, (d) size of the largest fused ring system, and the corresponding Kolmogorov-Smirnov distance to the training and test sets ($KS_{train}$ and $KS_{test}$, respectively) are reported. The same statistics from train and test sets (32,360 and 5,000 natural products, respectively) are reported for comparison. For each CLM and each metric, the best value is highlighted in boldface. All descriptors were computed on valid, unique, and novel SMILES.

| Metric | | S4 | LSTM | GPT | Training | Test |
|---|---|---|---|---|---|---|
| Syntax | Valid | **82,633 (81%)** | 76,264 (74%) | 70,117 (68%) | *n.a.* | *n.a.* |
| | Unique | **53,293 (52%)** | 51,326 (50%) | 50,487 (49%) | *n.a.* | *n.a.* |
| | Novel | 40,897 (40%) | **43,245 (42%)** | 43,168 (42%) | *n.a.* | *n.a.* |
| NP Likeness | Value | **1.6 ± 0.7** | 1.5 ± 0.7 | 1.5 ± 0.7 | 1.6 ± 0.7 | 1.6 ± 0.7 |
| | $KS_{train}$ | **4.03%** | 5.89% | 9.44% | 0.00% | 0.81% |
| | $KS_{test}$ | **4.51%** | 6.60% | 10.13% | 0.81% | 0.00% |
| No. $sp^3$ Carbons | Value | 42 ± 16 | 44 ± 17 | 43 ± 16 | 38 ± 16 | 37 ± 15 |
| | $KS_{train}$ | **13.96%** | 17.31% | 14.51% | 0.00% | 1.02% |
| | $KS_{test}$ | **14.08%** | 17.45% | 14.34% | 1.02% | 0.00% |
| Molecular Weight | Value | 1114 ± 315 | 1180 ± 360 | 1119 ± 307 | 1061 ± 295 | 1063 ± 290 |
| | $KS_{train}$ | **9.23%** | 16.97% | 11.02% | 0.00% | 1.40% |
| | $KS_{test}$ | **9.04%** | 16.67% | 10.75% | 1.40% | 0.00% |
| Size of the Largest Fused Ring System | Value | 5 ± 2 | 5 ± 2 | 5 ± 2 | 5 ± 2 | 5 ± 2 |
| | $KS_{train}$ | **8.05%** | 9.42% | 11.19% | 0.00% | 0.60% |
| | $KS_{test}$ | **7.93%** | 9.44% | 11.21% | 0.60% | 0.00% |

Murcko scaffolds (Bemis & Murcko, 1996). We then counted the number of obtained scaffold clusters, the higher, the better.

The models display similar trends with increasing $T$ values for all the analyzed factors across datasets, with varying magnitude (Figure 3). In general, the validity decreases with increasing temperature (as previously observed (Moret et al., 2020)), with the highest effect observed for GPT (median validity across training setups getting lower than 40%, Figure 3a).

Both S4 and LSTM show higher robustness than GPT to increasing temperature values (with LSTM performing slightly better for $T \geq 1.75$), suggesting that sequential generation can boost chemical space exploration. S4 outperforms LSTM in terms of rediscovery rate (Figure 3b), in agreement with our previous results on bioactivity (Figure 2). We also compute the exact rediscovery rate (identical molecular structure) and observe that no model can consistently generate held-out actives. When it comes to the diversity of the designs (Figure 3c), LSTM can generate the highest number of structurally unique scaffolds (median across datasets and setups: 6602, $T = 1.75$) and S4 is the close second-best model (6520, $T = 1.75$). While GPT obtains a suboptimal performance across the board, LSTM seems better for chemical space exploration when bioactivity is

not the main objective, while S4 can better capture bioactivity and preserve a good chemical space exploration at the same time, combining the strengths of the two benchmarks with its dual structure. These results confirm the promise of S4 when it comes to generating structurally diverse and bioactive drug-like molecules.

### 3.2. Designing natural products

S4 was further tested on more challenging molecular entities than drug-like molecules. To this end, we evaluated its capacity to design natural products (NPs), which are invaluable sources of inspiration for medicinal chemistry (Harvey et al., 2015; Atanasov et al., 2021). Compared to synthetic small molecules, NPs tend to possess more intricate molecular structures and ring systems, as well as a larger fraction of $sp^3$-hybridized carbon atoms and chiral centers (Lee & Schneider, 2001; Henkel et al., 1999; Chen et al., 2022). These characteristics introduce longer SMILES sequences on average, with more long-range dependencies, and make natural products a challenging test case for CLMs (Merk et al., 2018b; Ochiai et al., 2023).

We trained the CLMs on large natural products (32,360 SMILES strings with length $> 100$, chosen to complement the previous analysis) from the COlleCtion of Open Natural ProdUcTs (COCONUT) database (Sorokina et al.,

2021). We then used the CLMs to design 102,400 SMILES strings *de novo* and computed the fraction of valid, unique, and novel designs (Table 1). All CLMs can design natural products, with lower performance compared to drug-like molecules. S4 designs the highest number of valid molecules by approximately 6,000 to 12,000 molecules (7% to 13% better), and LSTM achieves the highest novelty by approximately 2,000 molecules (2%) over S4.

To further investigate the characteristics of the designs, we computed the natural-product likeness (Ertl et al., 2008), which captures how similar a molecule is to the chemical space covered by natural products in terms of its substructures (the higher the NP-likeness, the more similar). The novel designs of S4 have significantly higher (Mann-Whitney U test, $p < 0.01$) values of NP-likeness than the benchmarks, closer to the values of the training and test sets on average (Table 1). Moreover, the NP-likeness values better match the distribution of the COCONUT molecules in terms of Kolmogorov-Smirnov (KS) distance (Smirnov, 1939), which quantifies how much the cumulative distributions of two observations differ (between 0% and 100%; the lower, the closer the distributions).

We also evaluated the novel designs in terms of structural properties important for natural products (Lee & Schneider, 2001; Henkel et al., 1999; Chen et al., 2022), namely: the number of sp$^3$-hybridized carbon atoms, molecular weight, and size of the largest fused ring system. Here, S4 achieved the lowest KS distance to the training and test sets across the board, indicating that its designs match the training natural products best. These results confirm the ability of S4 to learn complex molecular properties for *de novo* design.

### 3.3. Prospective *de novo* design

Inspired by the effectiveness of S4 in capturing bioactivity and exploring the chemical space, we conducted a prospective *in silico* study to design inhibitors of MAPK1. The previously pre-trained S4 model was fine-tuned with the molecular strings of 68 manually-curated inhibitors from ChEMBL and 256K designs were generated. The designs were ranked and filtered via log-likelihood. The ten top-scoring molecules were considered for further evaluation using molecular dynamics simulations. We performed simulations also for the closest fine-tuning neighbor of the considered designs as a reference, . The absolute protein-ligand binding free energy (expressed as $\Delta G$; the lower the stronger the predicted binding) for molecules **1-16** was computed via Umbrella Sampling (Kästner, 2011) (Table 2).

8 out of 10 designs (except **1** and **5**) showed a high predicted affinity, with $\Delta G$ values ranging from $\Delta G = -10.3 \pm 0.6$ kcal/mol (**7**) to $\Delta G = -23 \pm 4$ kcal/mol (**2**). Interestingly, these affinities are comparable to or surpassing those of the closest active neighbor ($\Delta G = -9.1 \pm 0.8$ kcal/mol

*Table 2. Prospective design with S4. $\Delta G$ of the interaction (the lower, the better) was determined via molecular dynamic simulations. Mean and standard deviations of three runs are reported for de novo designs **1-10** and their nearest training inhibitor for comparison.*

| | S4 design | | Most similar training active |
|---|---|---|---|
| **ID** | $\Delta$**G** [kcal/mol] | **ID** | $\Delta$**G** [kcal/mol] |
| **1** | -5.6 ± 0.9 | **11** | -9.1 ± 0.8 |
| **2** | -23 ± 4 | **12** | -12 ± 2 |
| **3** | -19.6 ± 0.9 | **13** | -10.5 ± 0.7 |
| **4** | -13 ± 2 | **14** | -11 ± 3 |
| **5** | -7 ± 2 | **15** | -13 ± 2 |
| **6** | -11 ± 3 | **14** | -11 ± 3 |
| **7** | -10.3 ± 0.6 | **11** | -9.1 ± 0.8 |
| **8** | -11.2 ± 0.4 | **15** | -13 ± 2 |
| **9** | -17 ± 2 | **13** | -10.5 ± 0.7 |
| **10** | -15 ± 2 | **16** | -9.1 ± 0.2 |

to $\Delta G = -13 \pm 2$ kcal/mol); With 8 out of 10 designs predicted as bioactive on the intended target by molecular dynamics simulations, with comparable or higher predicted affinities than their closest fine-tuning molecules, these results further support the potential of S4 for *de novo* design.

## 4. Conclusions

This study pioneered the introduction of structured state-space sequence models (S4s) into chemical language modeling. The unique dual nature of S4s, involving convolution during training and recurrent generation, makes them particularly intriguing for *de novo* design with SMILES strings.

Our systematic analysis against GPT and LSTM on a variety of drug discovery tasks revealed S4's remarkable strengths: while recurrent generation (LSTM and S4) is superior in learning the chemical syntax and exploring diverse scaffolds, learning holistically on the entire SMILES sequence (GPT and S4) excels in capturing certain complex properties, like bioactivity. S4 with its dual nature, makes '*the best of both worlds*': it demonstrated comparable or better performance than LSTM in designing valid and diverse molecules, and systematically outperformed both benchmarks in capturing complex molecular properties. The application of S4 to MAPK1 inhibition, validated by MD simulations, further showcases its potential to design potent bioactive molecules.

Several aspects of S4 await to be explored in the molecular sciences, such as its potential with longer sequences (*e.g.*, macrocyclic peptides and protein sequences) and on additional molecular tasks (*e.g.*, organic reaction planning). We envision the relevance of S4 for molecule discovery to increase in the future, and to potentially replace widely established chemical language models like LSTM and GPT.

## Acknowledgements

The Python code and data to replicate and extend our study are available on GitHub at the following URL: https://github.com/molML/s4-for-de-novo-drug-design.

This research was co-funded by the European Union (ERC, ReMINDER, 101077879). Views and opinions expressed are however those of the author(s) only and do not necessarily reflect those of the European Union or the European Research Council. Neither the European Union nor the granting authority can be held responsible for them. The authors also acknowledge support from the Irene Curie Fellowship, the Centre for Living Technologies, and SURF (NWO grant EINF-5406). The authors thank Selen Parlar and the Molecular Machine Learning team (H. Brinkmann, C. Izquierdo-Lozano, M. Reksoprodjo, L. Rossen, Y.G. Nana Teukam, D. van Tilborg, L. van Weesep) for their feedback on the manuscript.

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
