# Supplementary Material

*Table S1. Designing drug-like molecules de novo with S4.* The results of LSTM and GPT models on the same tasks are also reported for comparison. Each model was trained on 1.9M SMILES strings from ChEMBL and used to generate 102,400 SMILES strings *de novo*. The number and percentage of valid, unique, and novel molecular designs are reported. The best value per metric is highlighted in boldface.

| Model | Valid | Unique | Novel |
|---|---|---|---|
| **S4** | **99,268 (97%)** | **98,712 (96%)** | **95,552 (93%)** |
| LSTM | 97,151 (95%) | 96,618 (94%) | 82,988 (81%) |
| GPT | 93,580 (91%) | 93,263 (91%) | 91,590 (89%) |

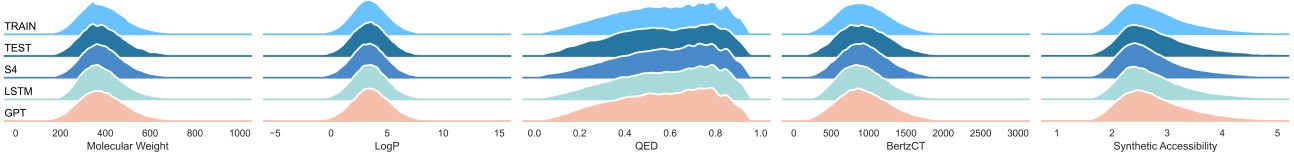

*Figure S1. Molecular descriptor distribution of designs after pre-training.* All models are sampled 102,400 designs and molecular weight, octanol-water partition coefficient (Log P) (Wildman & Crippen, 1999), quantitative estimate of drug-likeness (QED) (Bickerton et al., 2012), Bertz complexity (BertzCT) (Bertz, 1981), and synthetic accessibility (Ertl & Schuffenhauer, 2009) are computed.

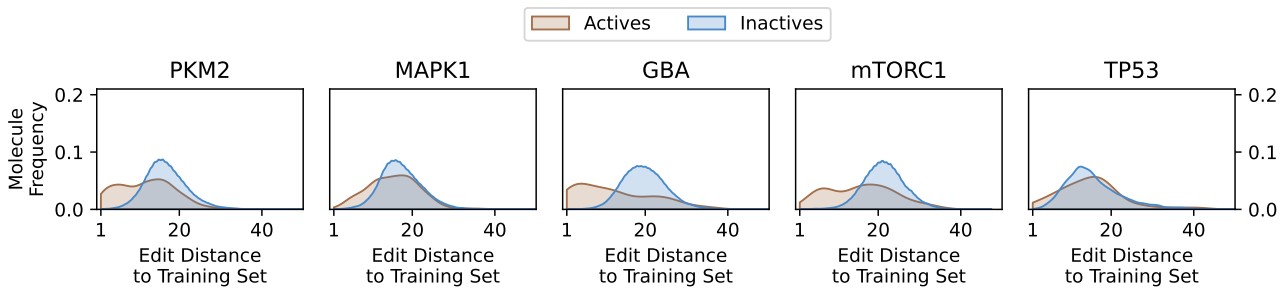

*Figure S2. Similarity of test set molecules to the training set.* Test sets across ten data splits are pooled together and the minimum edit distance (Levenshtein et al., 1966) of each test set molecule to the respective training set is computed per actives and ianctives.

*Table S2. Number of compounds used during the transfer-learnig phase.* Bioactive molecules were extracted from LIT-PCBA(Tran-Nguyen et al., 2020) database per each target and randomly divided into a training (80%), validation (10%) and test set (10%). Additionally, the test set contained 10,240 inactive molecules per target (chosen by random sampling). For TP53, all the inactive molecules available in the original dataset were considered.

| Dataset | Train | Valid. | Test Active | Test Inact. |
|---------|-------|--------|-------------|-------------|
| PKM2    | 436   | 54     | 56          | 10,240      |
| MAPK1   | 246   | 30     | 32          | 10,240      |
| GBA     | 132   | 16     | 18          | 10,240      |
| mTORC1  | 77    | 9      | 11          | 10,240      |
| TP53    | 44    | 10     | 10          | 3,301       |

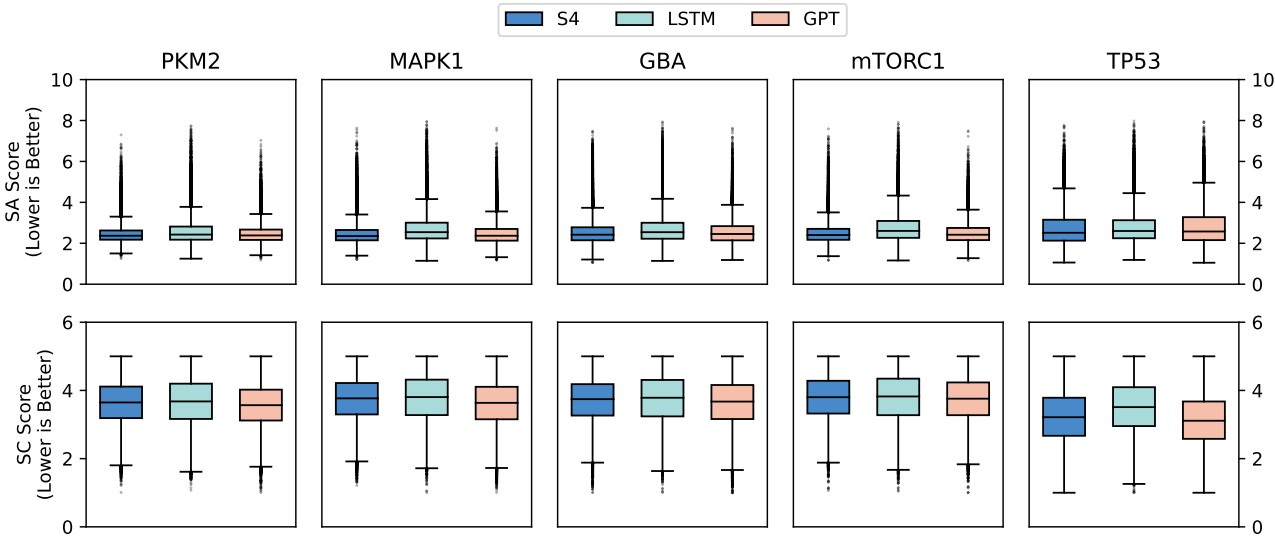

*Figure S3. Synthesizibility of the designs.* 10,240 designs were generated by each model per protein target and the synthetic accessibility (SA) score (Ertl & Schuffenhauer, 2009) and synthetic complexity (SC) score (Coley et al., 2018) were calculated.

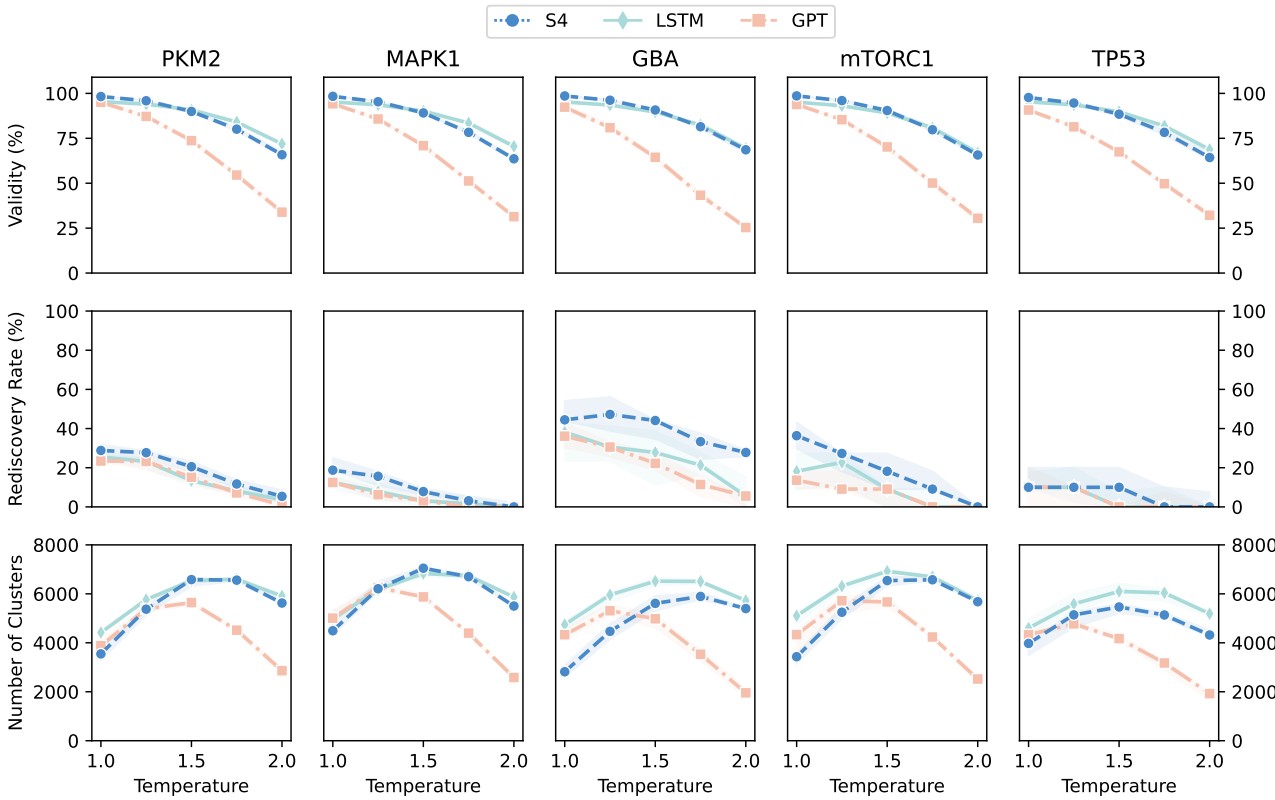

*Figure S4. Chemical space exploration per protein target.* The models were sampled in temperatures between 1 and 2 and validity, rediscovery rate (similarity above 60%), and number of scaffold clusters were computed.

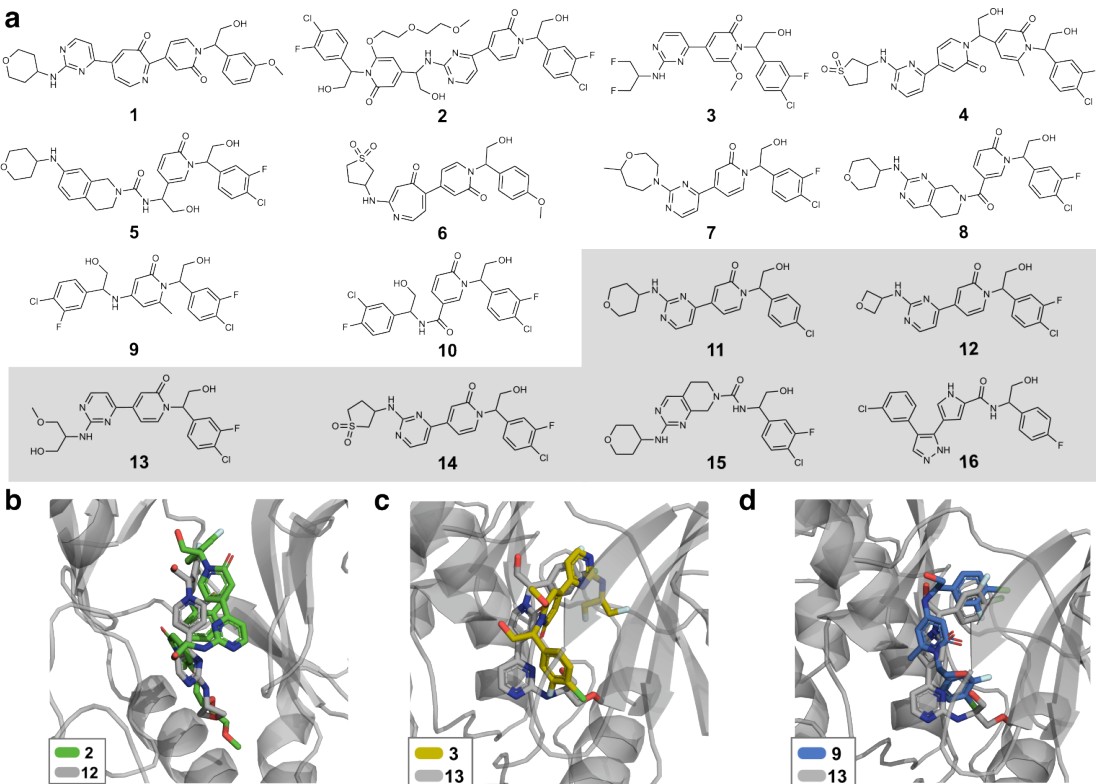

*Figure S5. Prospective de novo design of putative MAPK1 inhibitors with S4.* (**a**) Selected *de novo* designs (molecules **1** to **10**) for further characterization. For each *de novo* design, its most similar training MAPK1 inhibitor (as reported in Table 2) is depicted (compounds **11**-**16**, grey box). The ligand binding pose (obtained via Umbrella Sampling) of selected designs interacting with MAPK1 (PDB-ID=2Y9Q), in comparison with their most similar bioactive molecule from the fine-tuning set is also depicted: (**b**) Design **2** (green) compared with compound **12** (grey). (**c**), Design **3** (yellow) compared with compound **13** (grey). (**d**) Design **9** (blue) compared with compound **13** (grey).