# OpenReview forum: "Chemical Language Modeling with Structured State Spaces"
_ICML.cc/2024/Workshop/ML4LMS — ML4LMS Poster_

### Official Review · Reviewer_UUXs · 2024-06-12
**S4 models exhibit unique capabilities on chemical language modeling tasks**

**Rating:** 7
**Confidence:** 4

**Review:**

In this work the authors train a Structured State-Space Sequence (S4) model on Chembl and compare its capabilities for chemical language modeling to LSTM and transformer-based architectures.

Strengths
The authors’ approach to evaluating S4 on chemical language modeling is novel and appropriate for the capabilities of long context models. In particular, the SMILES design error and retrospective enrichment tasks are meaningful and informative. Focusing on design error rates as a way to compare the capabilities of different models to learn SMILES syntax makes sense, as does using likelihoods for ranking bioactivity. The paper is clearly written and well structured.

Weaknesses
Prospective design applications for chemical language models are extremely limited, and the sampling temperature sweep and in silico MD evaluation do not highlight the unique features of S4. The authors should contextualize these applications in particular with the approaches that they are not comparing against (nor do they need to if the focus is specifically on contrasting S4 with other language models): genetic algorithms, GNNs, enumeration and scoring, etc.

I recommend acceptance of the paper, and encourage the authors to continue developing out the more unique aspects of their work, focusing on the capabilities and limitations of long-context models compared to other chemical language models.

---

### Official Review · Reviewer_CakL · 2024-06-12
**Chemical Language Modeling with Structured State Spaces**

**Rating:** 7
**Confidence:** 3

**Review:**

The paper provides valuable insight into the applicability of S4 models in chemical language modelling. The authors perform benchmarking against LSTM and GPT models and demonstrate improved performance in terms of SMILES syntax and bioactivity. Additionally, a prospective in silico study for inhibitor design was conducted, which was validated using molecular dynamics simulations. The paper proves the relevance of S4 models for molecule discovery, however further studies are encouraged for a more stringent evaluation of the method, such as quantifying diversity using more insightful metrics (such as number of circles).

---

### Official Review · Reviewer_8Soo · 2024-06-12
**Could be helpful to have more comparisons**

**Rating:** 6
**Confidence:** 4

**Review:**

This work evaluates S4 for generative chemical tasks. While interesting, I think it could have been improved with better comparisons, and perhaps details/discussion. For example, what hyperparameter were used? Which GPT model was it? Which LSTM? There is often a conflict between optimizing for validity vs novelty. Also, how was chemical validity assessed?

Regarding validity, SELFIES have gained popularity, recently. Also, regression transformer would be interesting to compare against:

https://www.nature.com/articles/s42256-023-00639-z

AiS also:

https://jcheminf.biomedcentral.com/articles/10.1186/s13321-023-00725-9

I noted that they trained only on canonical SMILES. Augmentation has been found to be helpful:

https://arxiv.org/abs/1703.07076

What exactly is meant by bond assignment?

I believe that most of these issues should be fixable for a final version.